# Origin of charge transfer and enhanced electron–phonon coupling in single unit-cell FeSe films on SrTiO$_3$

Huimin Zhang[1], Ding Zhang[2,3], Xiaowei Lu[1,4], Chong Liu[2], Guanyu Zhou[2], Xucun Ma[2,3], Lili Wang[2,3], Peng Jiang[1,5], Qi-Kun Xue[2,3] & Xinhe Bao[1,5]

Interface charge transfer and electron–phonon coupling have been suggested to play a crucial role in the recently discovered high-temperature superconductivity of single unit-cell FeSe films on SrTiO$_3$. However, their origin remains elusive. Here, using ultraviolet photoemission spectroscopy and element-sensitive X-ray photoemission spectroscopy, we identify the strengthened Ti–O bond that contributes to the interface enhanced electron–phonon coupling and unveil the band bending at the FeSe/SrTiO$_3$ interface that leads to the charge transfer from SrTiO$_3$ to FeSe films. We also observe band renormalization that accompanies the onset of superconductivity. Our results not only provide valuable insights into the mechanism of the interface-enhanced superconductivity, but also point out a promising route toward designing novel superconductors in heterostructures with band bending-induced charge transfer and interfacial enhanced electron–phonon coupling.

[1] State Key Laboratory of Catalysis, CAS Center for Excellence in Nanoscience, Dalian Institute of Chemical Physics, Chinese Academy of Sciences, Dalian 116023, China. [2] Department of Physics, State Key Laboratory of Low-Dimensional Quantum Physics, Tsinghua University, Beijing 100084, China. [3] Collaborative Innovation Center of Quantum Matter, Beijing 100084, China. [4] University of Chinese Academy of Sciences, Beijing 100049, China. [5] Dalian National Laboratory for Clean Energy, Dalian Institute of Chemical Physics, Chinese Academy of Sciences, Dalian 116023, China. Huimin Zhang, Ding Zhang, and Xiaowei Lu contributed equally to this work. Correspondence and requests for materials should be addressed to L.W. (email: liliwang@mail.tsinghua.edu.cn) or to P.J. (email: pengjiang@dicp.ac.cn) or to X.B. (email: xhbao@dicp.ac.cn)

Heterostructure systems are a vibrant frontier for a number of emergent properties such as high-temperature superconductivity[1–3], Majorana fermion physics[4], topological Hall effect,[5] and ferroelectricity[6]. An epitome is the single unit-cell (1uc) FeSe on $SrTiO_3$ (STO) substrates where the superconducting temperature is significantly promoted[1–3, 7] to a value higher than 65 K[8, 9]. Previous experiments have indicated that charge transfer[7, 10, 11] and interface-enhanced electron–phonon coupling[1, 12–14] are two key factors for the observed high-temperature superconductivity in this system. The interface charge transfer removes the hole pockets in the Brillouin zone (BZ) center and enlarges the electron pockets around the BZ corner[7, 10–12], forming an electronic structure that resembles that of $A$Fe$_2$Se$_2$ ($A$ = K, Cs)[15]. Such an understanding has inspired the top–down doping scheme that effectively enhances superconductivity, as demonstrated recently in multilayer FeSe[13, 16–19] and Ba(Fe$_{1.94}$Co$_{0.06}$)$_2$As$_2$[20]. However, the origin of the charge transfer in FeSe/STO remains elusive experimentally, although oxygen vacancies[21, 22] and Se vacancies[23] have been theoretically considered. The situation becomes even less clear for the effect of electron–phonon coupling. Experimentally, the angle-resolved photoemission spectroscopy (ARPES) observation of shake-off bands suggests the coupling between FeSe electrons and optical phonons at the energy of ~100 meV in STO[12]. This phonon mode could be responsible for the further enhanced superconductivity in 1–3uc FeSe films on STO[14], compared with heavily electron-doped FeSe[18, 19, 24]. However, direct evidence for the enhancement of electron–phonon coupling as well as its correlation with superconductivity is still lacking. Elucidating the origin of interface charge transfer and electron–phonon coupling may stimulate another wave of searching for high-temperature superconductors by artificial construction of heterostructures.

In this study, we employ ultraviolet photoemission spectroscopy (UPS) and element-sensitive X-ray photoemission spectroscopy (XPS) to investigate the band alignment and electronic reconstruction of the 1uc-FeSe/STO heterostructure. We observe dramatic enhancement of the Ti–O bonding peak, suggestive of enhanced electron–phonon coupling, with annealing. We identify strong renormalization of Fe bands that accompanies the evolution of superconductivity. On the basis of work functions derived from UPS measurements and the band bending values extracted from XPS on core level states of individual elements, we construct the band profile across the FeSe/STO interface. We find that band bending of STO occurs close to the interface, giving rise to a substantial charge transfer to FeSe. These understandings suggest that high-temperature superconductivity may be achieved in a heterostructure consisting of both superconductive (SC) and dielectric layers. The dielectric layer not only enhances the charge density of the SC layer by charge transfer, but also provides the electron–phonon coupling with its intrinsic high-energy phonon mode.

## Results

**Surface morphology of 1uc-FeSe/STO.** Figure 1a shows the evolution of surface morphology of FeSe/STO after consecutive annealing at temperatures from 300 to 500 °C in ultra-high vacuum (UHV). At $T_{anneal}$ = 300 °C, the scanning tunneling microscopy (STM) image reveals atomically flat 1uc-FeSe film following the step-terrace structure of STO(001) surface. 2uc-FeSe islands (bright regions) always appear at lower-terrace

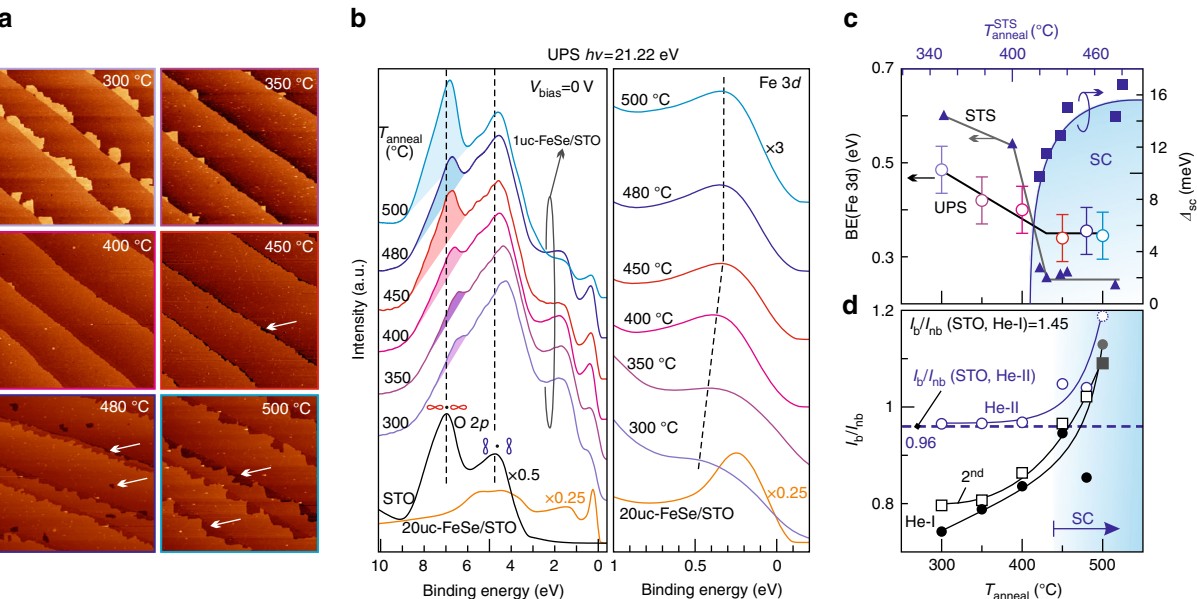

**Fig. 1** STM topography and valence band spectra of 1uc-FeSe/STO at different annealing stages. **a** STM images (500 × 500 nm$^2$) of 1uc-FeSe/STO annealed consecutively at 300, 350, 400, 450, 480, and 500 °C. *Arrows* indicate regions where the STO substrate is exposed. **b** Valence spectra of O 2$p$-and Fe 3$d$-derived features obtained by using the He-I light at 21.22 eV. All the spectra for FeSe/STO are normalized to the integrated intensity of O 2$p$ peaks. Curves are vertically offset for clarity. The *dashes* are a visual guide, marking the O 2$p$ features of STO. The *figure-of-eight patterns* represent the O 2$p$ orbitals pointing to (bonding) or avoiding (non-bonding) the Ti atom (*black dot*). *Shaded regions* highlight the Ti–O bonding related peak. **c** Correlation between annealing and superconductivity unveiled by the shift of the Fe 3$d$ peak observed both in UPS and low-temperature STS. The STS data were obtained from a separate system with the sample cooled to 4.6 K. The superconducting gap is defined as half of the distance between the two coherent peaks/kinks (Supplementary Fig. 2). The annealing temperatures for this system are marked on the top abscissa. Error bars were from the energy resolution of the measurement. **d** Intensity ratio between the bonding and non-bonding peaks as a function of annealing temperature probed by He-I (21.22 eV) (*solid circles* and *empty squares*) and He-II (40.81 eV) (*empty circles*) light sources. The *squares* represent data taken from a second sample following the same annealing process, which are offset by 0.1 for comparison

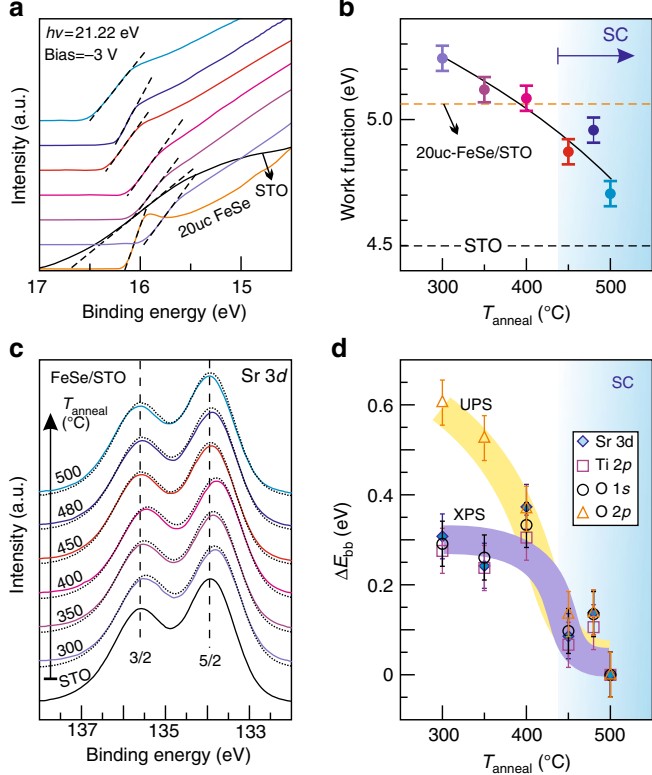

**Fig. 2** Work function of FeSe/STO and band bending in STO. **a** SECO edge obtained with the sample biased to −3 V and He-I light of 21.22 eV. Except for the bottom two curves, the curves are from 1uc-FeSe/STO at annealing stages in the same order as shown in Figure 1b. **b** Work function of 1uc-FeSe/STO as a function of annealing temperature. *Dashed horizontal lines* indicate the values of 20uc-FeSe/STO as well as a pristine STO, respectively. Error bars were from the energy resolution of the measurement. **c** Sr 3d core level spectra of pristine STO as well as STO after the growth of 1uc-FeSe and annealed at increasing temperatures. *Dotted curves* are fitted spectra described in Supplementary Fig. 9. *Dashed vertical lines* are guides to the eye. **d** Fitted band bending values at different annealing stages

step edges, indicating an ideal step-flow growth. With further annealing, 2uc-FeSe islands decompose and completely disappear at 400 °C. Due to the Se-rich growth condition (see Methods section), the 1uc-FeSe films at $T_{anneal} < 400$ °C contain abundant Se adatoms[25]. After sufficient annealing at 450 °C, extra Se adatoms desorb and the film becomes stoichiometric[25]. Decomposition occurs along the steps (*dark stripes* marked by the *white arrows* in Fig. 1a) but the FeSe coverage is still 99%. By increasing the annealing temperatures to 480 and 500 °C, the film coverage reduces to 92% and 86%, respectively. The remaining film is almost stoichiometric with minute Se deficiency[25].

**UPS spectra of 1uc-FeSe/STO.** Shown in Figure 1b are the normalized UPS spectra of 1uc-FeSe/STO with increasing annealing temperatures. For comparison, we also show the spectrum of a pristine STO substrate annealed at 1200 °C (*black curve*) and that for 20uc-FeSe/STO (*orange curve*). The spectrum of STO mainly consists of two peaks derived from O 2p orbitals in binding energy (BE) ranging from 3 to 9 eV. The higher energy peak at ~7 eV corresponds to Ti–O bonding state, whereas the lower energy one at ~5 eV is derived from O 2p non-bonding state[26]. The intensity drops abruptly at around 3 eV, from which

the valence band maximum (VBM) of 3.3 eV can be extracted by extrapolating a linear fit to zero intensity. Since the Fermi level here is almost aligned with the conduction band (Supplementary Fig. 1), its energy gap is roughly equal to the VBM value, i.e., $E_g$ (STO) = 3.3 eV.

Once 1uc-FeSe film is deposited, a small peak at ~0.5 eV and a broad hump at ~1.8 eV emerge in the gapped region of STO close to the Fermi level. They correspond to the Fe 3d derived bands[27, 28]. The right panel of Figure 1b provides a closer look of the Fe 3d feature at ~0.5 eV. With further annealing, this peak shifts to lower BE first from $T_{anneal}$ = 300 to 400 °C and stays nearly at the same BE from 450 to 500 °C. The intensity of the peak increases significantly from 350 to 400 °C, correlating with the desorption of extra Se. It then slightly drops at $T_{anneal}$ = 500 °C due to the decomposition of FeSe, as seen in Figure 1a. The spectral weight shift toward the Fermi level, instead of departing away from it as expected for electron doping, is characteristic of enhanced electron correlation. Coulomb interactions can renormalize a broad valence band into a peak and simultaneously pushes its spectral weight toward the Fermi level, as observed generally in iron-based superconductors[29, 30].

**Correlation between Fe 3d band and superconductivity.** The evolution of the Fe 3d band with annealing is consistent with low-temperature scanning tunneling spectra (STS) (Supplementary Fig. 2) and ARPES[7] results. Figure 1c summarizes the results obtained by UPS and STS, showing that the Fe 3d band shifts toward lower BE with increasing annealing temperature until it saturates at $T_{anneal} > 450$ °C. The low temperature STS further unveils that the superconducting gap opens up once the Fe 3d peak stops shifting and the magnitude of the gap gradually increases and reaches a maximum with further annealing (Supplementary Fig. 2). Previous ARPES investigations reveal consistent correlation between Fe 3d band position and emergence of superconductivity[7, 31]. We therefore correlate the saturated Fe 3d peak with the onset of superconductivity at low temperatures (*shaded region* in Fig. 1c). In the UPS spectra, the red shift of Fe 3d terminates at 400–450 °C. Therefore, both the UPS and the XPS spectra at $T_{anneal} \geq 450$ °C reflect the electronic structure of SC 1uc-FeSe/STO. In addition, the saturated Fe 3d peak situates at ~0.2 eV, further away from the Fermi level than that of a 20uc-FeSe (*orange curve* in right panel in Fig. 1b). This difference might stem from different doping level, as the 1uc-FeSe/STO is heavily electron-doped due to the band bending effect at the interface, which will be discussed later.

**Enhanced electron–phonon coupling with annealing.** With the 1uc-FeSe film epitaxially deposited and annealed, the O 2p peaks evolve systematically as well. As shown in the left panel of Figure 1b, the higher energy peak of O 2p level diminishes to a small bump just after the deposition of 1uc-FeSe, indicating a significant weakening of the Ti–O bond. This reduction may be caused by the extra Se atoms that intercalated at the interface of 1uc-FeSe/STO[32]. With annealing, however, the intensity of the Ti–O bonding peak gradually recovers (*shaded region* in Fig. 1b). Its peak position also exhibits a gradual blue shift. Meanwhile, the intensity of the non-bonding peak varies little, although its peak position exhibits the same blue shift as the Ti–O bonding peak does. Intriguingly, at $T_{anneal}$ = 450 °C, the Ti–O bonding peak pops up dramatically, corresponding closely to the SC stage. Figure 1d quantitatively draws the intensity ratio of the bonding peak over the non-bonding peak, i.e., $I_b/I_{nb}$ (Supplementary Note 1). A continuous rise can be seen from 300 °C to 450 °C. This trend has been confirmed on the second sample (*squares* in Fig. 1d). The shoot-up at 500 °C may partly arise from the

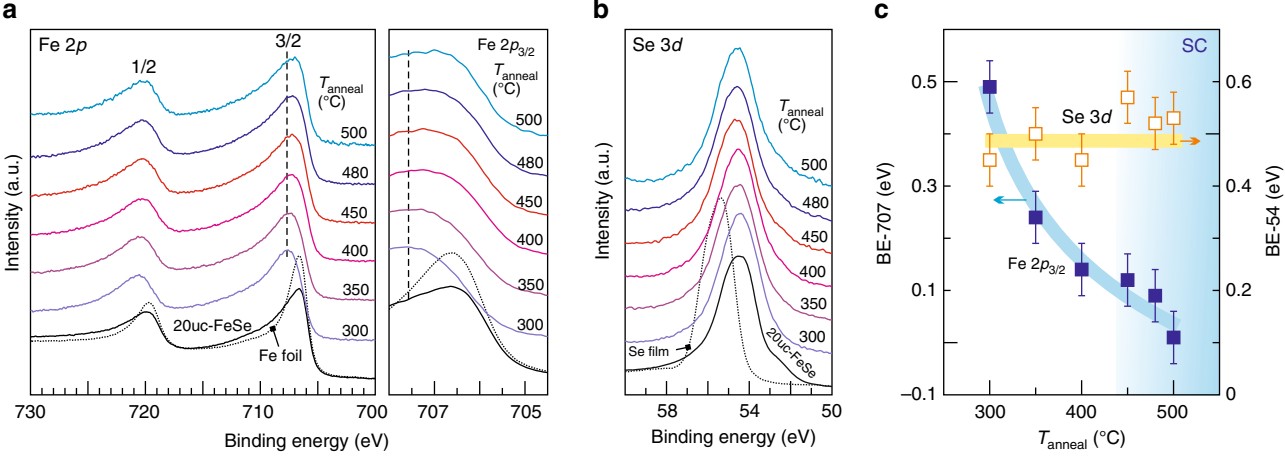

**Fig. 3** Fe $2p$ and Se $3d$ core-level spectra of 1uc-FeSe/STO. *Bottom curves* in the left panel of **a** (the main panel of **b**) are reference spectra of 20uc-FeSe/STO and a Fe foil (a 20 nm thick Se film). The Fe foil has been repeatedly sputtered and annealed to ensure the removal of oxidized layers. (**c**) Summary of the peak positions of Fe $2p_{3/2}$ and Se $3d$ at different annealing stages. The peak positions of Fe $2p_{3/2}$ are extracted from the Gaussian–Lorentzian fitting after subtracting the Shirley background (Supplementary Fig. 8). The peak positions of Se $3d$ are obtained from the corresponding energy values of the intensity maxima

exposed substrate, considering the fact that the initial STO has a much high $I_b/I_{nb}$ of 1.45. However, $I_b/I_{nb}$ still shows an increasing trend even after subtracting this contribution from STO (Supplementary Fig. 3). Furthermore, the spectra taken with He-II show that $I_b/I_{nb}$ of 1uc-FeSe/STO even exceeds the initial value taken from STO (0.96). In this situation, the sudden increasing ratio cannot be attributed to the exposed STO region at all. We therefore interpret the increasing $I_b/I_{nb}$ from 300 °C up to 500 °C as a manifestation of the strengthened Ti–O bonds in STO beneath FeSe in comparison to the as-grown state. Since the mean free path of electrons in this ionization energy range is about 1 nm[33], the intensity ratio we address here involves mostly the surface bonds. Its close correspondence with superconductivity suggests that the surface Ti–O bonds play a pivotal role in the high-temperature superconductivity of FeSe/STO. Especially, phonons supported by these bonds in STO may interact with electrons in FeSe and raise the transition temperature. For example, the optical phonon mode responsible for the replica bands seen in ARPES[12] propagates along the Ti–O directions[34]. At the as-grown state, the abundant Se adatoms may largely screen the dipole field generated by the F–K phonon modes in STO[14] and the substrate-mediated electron–phonon coupling poorly manifests itself. With annealing, Se adatoms desorb and the electric field can penetrate to the FeSe such that the substrate-mediated electron–phonon coupling becomes stronger. For $T_{anneal} > 450$ °C, the difference between the Ti–O bonding peak and non-bonding peak keeps growing although the renormalization effect is already saturated. This contrast seems to suggest that the electron–phonon coupling becomes stronger and such a mechanism further widens the superconducting gap, beyond doping alone.

**Band bending**. We now address the energy band profile across the FeSe/STO heterostructure. Figure 2a shows the cut-off edge of the UPS spectra with the sample biased to −3 V. They reflect the deepest electrons residing in the valence bands that can be excited by the 21.22 eV photons. The work function of sample can be extracted by using the formula: $\phi = h\nu - $ SECO, in which the secondary electron cut-off (SECO) is obtained from a linear extrapolation (*dashed lines* in Fig. 2a). Thus, we obtain $\phi$ (STO) = 4.5 eV, $\phi$ (20uc-FeSe/STO) = 5.1 eV and $\phi$ (1uc-FeSe/STO) =

4.8–5.2 eV, as summarized in Figure 2b. The obtained work function of STO is consistent with previous reports[35]. When FeSe is deposited onto STO, electrons in STO transfer to FeSe to compensate for the large difference in work functions. Such an electron transfer process terminates when the Fermi levels of both systems align together to reach thermal equilibrium. Consequently, the energy bands in STO bend upward to accommodate such an alignment. This bending is captured by the shift of the O $2p$-derived features to lower BE once FeSe is deposited (Fig. 1b). With further annealing, however, the peaks shift back toward higher BE. This weakening of band bending is in line with the reduction of $\phi$ (1uc-FeSe/STO) (Fig. 2b): consecutive annealing promotes the electron density in FeSe and pushes up the Fermi level.

In order to understand the band bending quantitatively, we turn to the XPS data. The XPS measures the core level states of individual elements, which are less entangled than the bands studied by UPS. Consistent with the behavior of O $2p$ level shown by the UPS (Fig. 1b), the levels for each component in STO, i.e., Sr $3d$, Ti $2p$, and O $1s$, exhibit a slight red shift after FeSe deposition and then shift back to higher BE with further annealing (Supplementary Fig. 4). The apparent shift is, however, much subtler due to the relatively deeper penetration depth of the X-ray, where the bulk bands contribute a strong background signal. To quantify the band bending, we employ a fitting method (Supplementary Note 2) by summing up the layer-by-layer contributions. The top layer has a larger spectral weight shift due to a larger band bending while the layer deep in the bulk has a negligible energy shift[36]. We employ the up-to-date structural information with double TiO$_2$ layers of STO at the interface[32, 37], i.e., TiO$_2$/TiO$_2$/SrO/TiO$_2$/SrO.... Switching to the situation of a single surface TiO$_2$ layer affects the fitted results but the difference falls within the error bar (Supplementary Fig. 5). Figure 2c displays the spectra of Sr $3d$, where *solid curves* correspond to the spectra normalized by the integrated intensity around the peaks and the *dotted curves* exemplify the fitted spectra with the best-fitted band bending values. The extracted band bending values from Sr $3d$, Ti $2p$, and O $1s$ are summarized in Figure 2d. The STO bands bend upwards for about 0.3 eV in the initial non-superconductive (NS) stage of FeSe/STO. The band bending becomes weaker with annealing and drops to around 0.1 eV in the SC stage. At 500 °C, the band bending

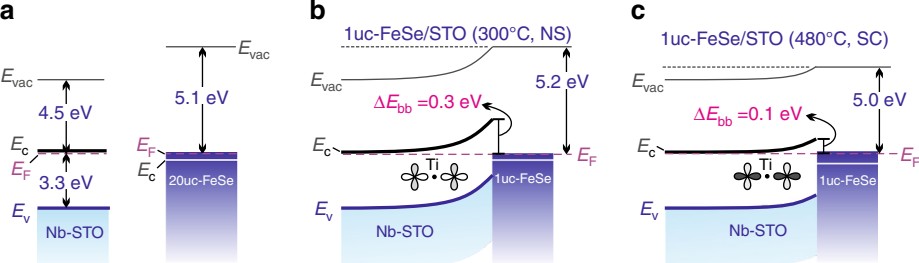

**Fig. 4** Band alignment of FeSe and SrTiO₃. **a** Energy bands of Nb-doped STO and 20uc-FeSe separately. **b, c** Energy band profile across the FeSe/STO heterostructure at the NS and SC stages, respectively. The *double figure-of-eight patterns* represent the O 2p orbitals. *Dark color* highlights the bonding strength

becomes challenging to resolve with our instrument. The exposed STO may be one factor that blurs the extracted bending. For a modulation-doped semiconductor heterostructure, increasing the remote doping concentration introduces more charge carriers into the interface. Concomitantly, it sharpens the band bending since the conduction band of the doped region becomes closer to the Fermi level (Supplementary Fig. 6). Here we observe just the opposite: the increase of electron density in FeSe correlates with the weakening of band bending in STO. It suggests that vacuum annealing here works rather differently than introducing remote dopants (Nb, for instance), thus changing the Fermi level in the bulk STO. One may speculate that annealing creates oxygen vacancies in STO close to the interface, which may reduce the band bending and provide electrons directly to FeSe. However, such a scenario fails to explain the same bending values obtained from the spectra of all three elements: Sr, Ti, and O (Fig. 2c and Supplementary Fig. 4). We therefore attribute the increased electron density with post-annealing mainly to the stoichiometric effect of FeSe. That is, band bending is responsible for the interface charge transfer from STO to FeSe, and FeSe itself gains a higher electron density with the desorption of extra Se[25, 38]. In addition, we include the fitting results of O 2p peak obtained by UPS with the 40.81 eV He-II light (Supplementary Fig. 7) in Figure 2d as well. At $T_{anneal}$ = 300 and 350 °C, the fitted values obtained from the UPS data are ~0.2–0.3 eV larger than the ones from XPS results. We attribute the sharp contrast at these stages to abundant Se atoms at the interface. They contribute to the same energy window of the O 2p valence bands probed by UPS[29]. The fitted values from XPS and UPS become almost overlapping when extra Se atoms are removed at $T_{anneal}$ > 350 °C.

**Core level spectra of Fe and Se**. Displayed in Figure 3a, b are the Fe 2p and Se 3d spectra normalized by the integrated intensity from 705 to 725 eV and 51 to 58 eV, respectively. The Fe 2p spectra consist of $2p_{3/2}$ and $2p_{1/2}$ peaks due to spin-orbit interactions. As a reference, Fe foil corresponds to zero valence state: Fe⁰ (*dotted black curve* in Fig. 3a). The resemblance of the spectrum from the 20uc-FeSe/STO (*solid black curve* in Fig. 3a) to that of Fe⁰ reflects the delocalized nature of electrons in the Fe 3d band, as has been observed previously in bulk FeSe[29] and CeFeAsO₀.₈₉F₀.₁₁[39]. In the case of 1uc-FeSe/STO, however, the Fe $2p_{3/2}$ and $2p_{1/2}$ peaks become much broader and situate at BE higher than that of 20uc-FeSe/STO. We surmise that a fraction of the spectral weight now arises from the more localized valence states of Fe, i.e., Fe²⁺. This enhanced localization suggests that electron correlation in 1uc-FeSe/STO is stronger than that in bulk FeSe, in agreement with previous ARPES results[31]. The Se 3d doublets (5/2 and 3/2 constitute one peak) of 1uc-FeSe/STO and 20uc-FeSe/STO always sit at lower energies than that of

a Se film (Fig. 3b). The bump at 52.5 eV in the 20uc-FeSe/STO spectrum stems from Fe 3p[40]. With annealing, the Fe 2p peak shifts to lower energies (right panel of Fig. 3a) while its peak width slightly narrows (Supplementary Fig. 8). In contrast, the Se 3d peak stays unaltered (Fig. 3b). Such distinctly different behaviors are further captured in Figure 3c. Similar to the evolution of Fe 3d peak revealed by UPS (Fig. 1c), the change in Fe 2p peak also saturates once entering the SC regime, providing evidence for the correlation effects on the core level states. This finding expands the previous ARPES studies focusing on the energy range close to the Fermi level[30, 31]. It is worth pointing out that the observed behavior of the core level states is markedly different from the stubborn Fe valence state in Co-doped BaFe₂As₂ and SrFe₂As₂. There[41, 42], neither appreciable alteration in shape nor energy position for Fe element was observed. The orbital-selective renormalization of Fe bands may be a unique feature of iron chalcogenide superconductors[30, 31].

## Discussion

On the basis of the above results, we highlight our findings in Figure 4. First, electron doping in FeSe stems from two mechanisms. On the one hand, since FeSe has a larger work function than Nb-doped STO (Fig. 4a), band bending occurs on the STO side of the FeSe/STO heterostructure. The upward bending in STO constitutes a major charge transfer to FeSe (Fig. 4b). On the other hand, with the chemical environment of FeSe varies from Se rich to stoichiometric with annealing, its Fermi level moves upward and the electron density further increases[7, 25, 38]. Hence, the barrier across the FeSe/STO junction becomes smaller with annealing (from Fig. 4b to Fig. 4c). Second, the interface charge transfer induces orbital-selective strong renormalization of Fe bands (Figs. 1b and 3a), which could trigger a new superconducting phase with superconducting transition temperature around 40 K and superconducting gap of 10 meV as achieved by doping alone[18, 31, 43]. Last but yet important, apart from the electron-doping effect, the Ti–O bond becomes strengthened with annealing in comparison to the as-grown state (double figure-of-eight in Fig. 4c). It reflects the enhancement of the substrate-mediated electron–phonon coupling, which further promotes the superconductivity. Such a mechanism explains the observation that ultra-thin FeSe films on STO always exhibit larger superconducting gaps, which exponentially decay with increasing thickness up to 3uc[13, 19] than other heavily electron-doped FeSe[18, 31, 43] systems. It reinstates the vital role of Ti–O bond, as similarly large superconducting gap and shake-off bands were recently observed in 1uc-FeSe films on TiO₂ substrates[44, 45]. This work sheds light on choosing the appropriate material for realizing interface-enhanced superconductivity. Apart from the demanding growth

conditions such as matching the lattices, the substrate should have a lower (higher) work function to allow for charge transfer that promotes the charge carrier density in the n-type (p-type) charge-transporting layer and have high-energy phonon modes to further enhance the pairing strength.

## Methods

**FeSe film preparation**. The experiments were performed in an UHV system equipped with STM, XPS, UPS, and molecular beam epitaxy (MBE), which allows in situ growth, annealing, and characterization. The substrate is Nb-doped STO(001) (0.7 wt%), which exhibits a clean and atomically flat surface with regular steps after being annealed up to 1200 °C. The 1uc-FeSe film was prepared by co-evaporating high purity Fe (99.995%) and Se (99.9999%) under the Se-rich condition at a substrate temperature of 300 °C. All samples were annealed for 2 h at each temperature (350, 400, 450, 480, and 500 °C).

**UPS and XPS measurements**. In situ STM, XPS, and UPS measurements were performed at room temperature after each annealing treatment to record the evolution of morphology, chemical bonding, and work function, respectively. XPS spectra were recorded using Mg $K_\alpha$ radiation ($hv = 1253.6$ eV) with a pass energy of 30 eV and an emission angle of 53° with respect to the sample plane. All spectra were calibrated using the reference spectra of a Pt foil. Peak positions for Fe were determined by fitting with Gaussian–Lorentzian functions after subtracting a Shirley nonlinear sigmoid-type baseline, while those for Se were evaluated by the peak maxima. For UPS measurements, both He I and He II light sources ($hv = 21.22$ and 40.81 eV) were used with an emission angle of 61° with respect to the sample plane. The Fe foil used as a reference was sputtered and annealed in UHV for several cycles to remove contamination and oxides. The peak position for Fe $2p_{3/2}$ was at 706.56 eV. A Se film was obtained by depositing Se on STO substrates at room temperature for 1.5 h, which resulted in a nominal film thickness of 20 nm. The peak maximum of Se $3d$ was at 55.35 eV. Error bars of the peak positions were estimated by considering the energy resolution of the measurement (0.05 eV).

**LT-STM measurements**. Low-temperature STM/STS measurements were carried out in a separate MBE-STM system. A polycrystalline PtIr tip was used, and the STM stage was cooled to 4.6 K. The STS was acquired by using a lock-in technique with a bias modulation of 0.5 mV at 437 Hz.

**Data availability**. The data that support the findings of this study are available from the corresponding authors on request.

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

## Acknowledgements

We thank Jinfeng Jia, Canli Song, Wei Li, Lexian Yang, Keith Gilmore, and Yi Lu for fruitful discussion. P.J. acknowledges financial support from the National Natural Science Foundation of China (NSFC) (grant no. 51290272) and Open Research Fund Program of the State Key Laboratory of Low-Dimensional Quantum Physics of Tsinghua University (grant no. KF201511). The work in Tsinghua University is supported by NSFC (grant nos. 91421312, 11574174, and 21573121).

## Author contributions

P.J., L.W., Q.-K.X., and X.B. designed the research. H.Z. and X.L. carried out the MBE thin-film growth and in situ measurements with the assistance of G.Z.. C.L. carried out the MBE thin-film growth and low-temperature STS measurements. H.Z., D.Z., P.J., L.W., and Q.-K.X. analyzed the data and wrote the paper. All authors discussed the results and commented on the manuscript.

## Additional information

**Competing interests:** The authors declare no competing financial interests.

