## [Peer Review File · Nature Communications]

Reviewers' comments:

Reviewer #1 (Remarks to the Author):

The authors perform ultraviolet and x-ray photoemission spectroscopy and obtain tunneling spectra to study the position in energy of electronic states attributed to the individual kinds of atoms in the system Fe, Se, Ti, Sr and O. After giving a reference for the energy level on the O p states of pristine SrTiO₃ (STO), the authors discuss the position of the presumed Fe d states that appear once the 1 monolayer FeSe surface is placed on the substrate as a function of annealing steps at temperatures T= 300,350,400,450 and 480C.

They find that a peak in the spectral weight associated to Fe d-orbitals moves closer to the Fermi level and for T>=450 saturates upon annealing and which is in agreement with previous work. They attribute this effect to increasing correlations. Comparison with the 20 unit cell (uc) film reveals that the saturated peak at ~0.2 eV of the 1 uc film is further away from the Fermi level which the authors interpret as electron doping.

The authors further discuss the O-p level separated into bonding Ti-O bonding and non-bonding part. While an initial weakening of the bonding part is attributed to intercalated Se, it is found that upon annealing the bonding peak is enhanced as compared to the non-binding one.

The authors measure the work function and find that annealing reduces the 1 uc FeSe/STO workfunction from an initially higher value than 20 uc FeSe to much below that approaching the value of pristine STO. In line with this observation, they find that the O-p level shift from an initially low energy value to towards the Fermi level upon annealing. The same observation is made using x-rays on core level electronic states Sr 3d, Ti 2p and O 1s where the authors note that this method averages multiple layers.

The authors observe that while charge is transferred to FeSe upon annealing, the energies of all considered states in STO move towards the Fermi level. The trend is consistent in XPS and UPS and the amount eventually agrees between the methods upon reaching 400C which suggests that range of charge depletion becomes smaller which is counter intuitive. The authors explain this by attributing the increase of charge in the FeSe layer to the desorption of extra Se atoms at higher temperatures.

Now, the authors focus on the Fe and Se core level states. They speculate that from the broadening of Fe 2p states in energy, which now overlap with the localized Fe-d orbitals, there could be hybridization of these two kinds of orbitals. They claim that such an enhanced localization is indicative of strong correlations. The authors track the position of the Fe 2p peak and find that it move towards the Fermi level upon annealing while the Se 3d level remains inert. From this they conclude that the doped electrons are mostly transferred to the Fe layer.

Finally, the authors summarize and claim that from the alignment of the bands to more equal Fermi level upon annealing it is to be expected that the electronic wave functions penetrate deeper into the substrate. Together with the strengthened Ti-O bond they infer that coupling to such a vibration will enhance the T_c to the higher values of the FeSe/STO interface as compared to the electronically similar electron doped bulk FeSe. They point out that while states do not shift in energy upon reaching the superconducting phase, as T_c increases the Ti-O becomes stronger and stronger.

I think this paper is important for specialists working on FeSe on STO but makes claims that are not supported by facts. Because the data presented for the evolution of the particular bands under annealing is quite interesting, I recommend it for publication in Nature communications after considering my revisions.

1.

Abstract: "...but also point out an explicit route for designing novel high temperature superconductors."

I don't see any route to novel superconductors described in this paper except for a vague sentence in the end. I think this claim has to be dropped or such a route actually explained.

2.

Page 1 first paragraph:

"This phonon modes decays into FeSe ..."

I am not aware that the phonon modes under discussion involve a vibration of FeSe. In any case it is unclear what "decay" means in this context and this statement has to be corrected.

3.

Page 1 second paragraph:

"This band diagram ... the model proposed recently for cuprate high temperature superconductors."

I feel this sentence is unclear. The authors mention "the model" but do not explain in what sense the results are similar and thus this statement is not helpful. I think the authors should clarify in what sense a "similarity" is present or remove the sentence.

4. Page 4 second paragraph:

"On the other hand, the Ti-O bond could correspond to the dispersionless phonon mode at ~ 100 meV which generates the replica bands seen in ARPES, ..."

Which atoms vibrate due to this mode is well studied in the literature. What is important is less the dispersion and more the coupling which must be peaked at $q=0$ to explain the replica bands. Statements regarding the electron-phonon coupling in general are very vague in the entire manuscript and need to be clarified or removed.

a) "the permeation of electrons of FeSe into STO" as the authors claim is not reasoned except for a vague statement that the Fermi energies align which could or could not mean that electronic states hybridize.

b) The authors fail to recognize that scenario to explain the replica bands via electron-phonon coupling was built exactly on the idea that FeSe states do not penetrate.

In summary, if the authors want to make claims in that direction, I feel they should review the literature properly and introduce their explanation of how the stronger bond leads to a significant increase in coupling.

5. Page 5 last paragraph

This paragraph is unclear and should be made more precise.

a) The notation $p_{1/2}$ and $p_{3/2}$ and $5/2$ is not introduced

b) I failed to understand the logic of the two sentences "The Fe 2p spectrum ... " until " as has been observed previously in bulk FeSe and CeFeAsO"

c) "... those of Fe in the zero valence state ..." The authors should explain what is meant by "the zero" in this case.

6. Page 6 first paragraph:

"Screening around Fe ... enhances the correlation effects of Fe bands only. A recent calculation ... of the balance states"

Before this sentence the authors are referring to the Fe p state and I believe they want to make the claim that the 2p state moves up in energy because of correlations that increase the effective mass and not because the Fermi level is reduced. However, they also say that from this peak the doped electrons concentrate on the Fe layer and I don't see who these conclusions are arrived at. I think the entire paragraph should be reformulated to state clearly 1) what is measured 2) the hypothesis and 3) why the data allows for this interpretation.

7. Page 6 second paragraph:

"... the barrier across the FeSe/STO junction becomes smaller and the wavefunction of FeSe electrons spread deeper into STO ..."

While I can understand this argument from a particle-in-a box like argument, the Fermi level electrons are mostly Fe which is relatively far away from STO. Most importantly, for electron phonon coupling (and this is where the authors are going for) the vibration of an atom has to induce a potential variation for the particular electronic state in question. Even if Fe electrons tunnel into the STO more with annealing, it is unclear why this would increase the electron phonon coupling because the Fe and the Ti-O bond do not hybridize.

The model put forward by the original forward-scattering nature paper is in disagreement with this picture, because they have an effective distance d of the polar mode in STO and the Fe layer and a large d ensures the sharpness of the coupling parameter $g(q) \sim \delta(q)$ required to explain the replica bands.

Thus, the logic the authors present in this paragraph is not sound. If they want to make a statement regarding electron-phonon coupling they should embed their picture and data into the relevant literature.

For example, following their picture of increased tunneling of Fe into STO, I would have to expect that the electron-phonon coupling becomes broader in q which is not in line with the observed replica bands and in contrast to the claim of the paper.

8. Page 6 and 7, last paragraph:

"Our findings here echo ..."

The authors want to make the reader aware of their picture of the cuprates, but I feel the sentence is out of place, because it is unclear why a similar model in an unrelated material should help to understand the present case. If they want to keep a similar sentence, the authors should explain clearly in what sense the cuprates are related to FeSe on STO and thus, why the analogy helps to understand the present material.

9. Page 7: "Novel superconductors may be ..."

The authors conclude with two sentences how novel superconductors could be constructed. Their suggestions are vague and largely based on the picture presented with the enhanced penetration of Fe electrons into the STO. As mentioned before, this picture is not in line with the literature and also not argued carefully.

Reviewer #2 (Remarks to the Author):

In the present work, the authors report STM, UPS, and XPS results on monolayer FeSe/STO. The authors performed very careful UPS and XPS measurements on the well characterized FeSe/STO with superconducting gap confirmed by STS. Such conventional UPS and XPS on the well-defined samples can be more useful than high-resolution ARPES for understanding of interface electronic structure. Indeed, this is the case in the present study.

The evolution of Fe 3d band with annealing is very interesting and consistent with the emergence of superconductivity. Also the decrease of band bending and work function with annealing is clearly shown by the systematic UPS and XPS data and supporting the band profile model. However, I have a serious concern about the interpretation on the popup in the O 2p band of the 500oC annealed sample. The authors assigned the popup to strengthened Ti-O bonding and argued that "the Ti-O bonding could correspond to the dispersionless phonon mode at about 100 meV which generates the replica bands seen in ARPES." This assignment and the argument are too speculative. I suspect that the popup in the O 2p band would be related to a kind of disorder in the topmost FeSe layer since the intensity of Fe 3d band is reduced in the 500oC annealed sample (Fig. 1b) and would be consistent with the increase of exposed STO area in Fig. 1a. I recommend the authors to show Fe 2p XPS spectra (as well as Se 3d and Sr 3d) for the 500oC annealed sample in Figs. 2 and 3 and to reexamine the electronic states of the 500oC annealed sample as well as the origin of the popup in the O 2p band of the 500oC annealed sample.

As a minor point, I recommend the authors to check English of the manuscript. There are some misprints (for example, "vidual" in the caption of Fig.1).

In conclusion, although the present work is potentially important and may deserve publication in Nature Communications, I cannot recommend publication of this manuscript in the present form.

Reviewer #3 (Remarks to the Author):

The manuscript "Origin of charge transfer and enhanced electron-phonon coupling in single unit-cell FeSe films on SrTiO₃" by Zhang et al. presents UPS/XPS studies on the high T_c superconductor 1uc FeSe/STO. The origin of the enhanced superconductivity in this system has been a popular research topic, because the answer might help to guide the design and research of other interfacial superconductors. This manuscript measured the enhanced Ti-O bonding and the evolution of band bending during the annealing process from NS to S states. It also discussed the effect on charge transfer and e-p coupling. The paper is well written and seems scientifically sound. I want authors to clarify the following questions and provide some discussions in the revised version before the manuscript can be published in Nature Communications.

1. Please provide error bars for Figures 1 and 2.
2. The authors show that the Ti-O bonding becomes weak once the FeSe deposited and stronger during the annealing process. What's the $I_{\text{bond}} / I_{\text{nonbond}}$ ratio for annealed STO substrates? If the bonding in FeSe/STO after annealing is similar to STO, the authors should be careful about claiming the bonding is "strengthened" (compared to STO). The deposition of FeSe may weaken the Ti-O bond on the surface at the beginning, and later it becomes bulk-like during the annealing.
3. Several papers have shown the double TiO₂ layer on STO surface (2D Mater., 024002, 2016 (Ref. 34 in the text) and PRB, 195303, 2016 (not in Ref.), and so on). The measured Ti-O bond length is not obviously smaller in the 1st TiO₂ layer. Also the bottom Se-Ti distance is bigger than expected, which is shown to be the opposite in calculations PRB, 220503, 2013 (Ref. 21 in the text) as the result of enhanced coupling of FeSe and STO. Can the authors provide some discussions?

Reviewer #1:

I think this paper is important for specialists working on FeSe on STO but makes claims that are not supported by facts. Because the data presented for the evolution of the particular bands under annealing is quite interesting, I recommend it for publication in Nature communications after considering my revisions.

We thank the referee for providing positive and constructive feedback. We have taken significant revisions based on the referee's suggestions. Below we list them according to the points raised one-by-one:

1. Abstract: "..., but also point out an explicit route for designing novel high temperature superconductors." I don't see any route to novel superconductors described in this paper except for a vague sentence in the end. I think this claim has to be dropped or such a route actually explained.

We have replaced this sentence by a more specific one based on our experimental observations: "Our results not only provide valuable insights into the mechanism of the interface-enhanced superconductivity, but also point out a promising route towards designing novel superconductors in heterostructures with band-bending induced charge transfer and interfacial enhanced electron-phonon coupling."

We have also rewritten the last paragraph and provide a more explicit suggestion about the choice of materials: "This work sheds light on choosing the appropriate material for realizing interface enhanced superconductivity. Apart from the demanding growth conditions such as matching the lattices, the substrate should have a lower (higher) work function to allow for charge transfer that promotes the charge carrier density in the n-type (p-type) charge-transporting layer and have high energy phonon modes to further enhance the pairing strength."

2. Page 1 first paragraph: "This phonon modes decays into FeSe ..."

I am not aware that the phonon modes under discussion involve a vibration of FeSe. In any case it is unclear what "decay" means in this context and this statement has to be corrected.

We thank the referee for pointing out this incorrect use of word. We have removed "decay into FeSe" in this sentence. From the cited reference [14], it is the electric field from STO that

penetrates into FeSe. Such a mechanism is believed to produce the substrate mediated electron-phonon coupling. Phonons in STO, as the referee pointed out, do not decay into FeSe.

3. Page 1 second paragraph: “This band diagram ... the model proposed recently for cuprate high temperature superconductors.” I feel this sentence is unclear. The authors mention “the model” but do not explain in what sense the results are similar and thus this statement is not helpful. I think the authors should clarify in what sense a “similarity” is present or remove the sentence.

We have rewritten this section and removed the discussion about cuprates. We have added the sentence: “These understandings suggest that high temperature superconductivity may be achieved in a heterostructure consisting of both superconductive and dielectric layers, where the dielectric layer not only enhances the charge density of superconductive layer by charge transfer, but also strengthens the electron-phonon coupling at the interface due to its intrinsic high energy phonon mode.”.

4. Page 4 second paragraph: “On the other hand, the Ti-O bond could correspond to the dispersionless phonon mode at ~100 meV which generates the replica bands seen in ARPES, ...” Which atoms vibrate due to this mode is well studied in the literature. What is important is less the dispersion and more the coupling which must be peaked at $q=0$ to explain the replica bands. Statements regarding the electron-phonon coupling in general are very vague in the entire manuscript and need to be clarified or removed.

a) “the permeation of electrons of FeSe into STO” as the authors claim is not reasoned except for a vague statement that the Fermi energies align which could or could not mean that electronic states hybridize.

b) The authors fail to recognize that scenario to explain the replica bands via electron-phonon coupling was built exactly on the idea that FeSe states do not penetrate.

In summary, if the authors want to make claims in that direction, I feel they should review the literature properly and introduce the explanation of how the stronger bond leads to a significant increase in coupling.

We thank the referee for correcting us on this very important point. Our experimental observation is that the Ti-O bond becomes stronger with annealing, which correlates with the

onset of superconductivity. We speculate that this strengthened bond gives rise to the enhanced electron-phonon coupling, which in turn promotes superconductivity. The original statement of ours was a little bit confusing. We have removed the speculation about electron permeation here and also the related discussion throughout the paper.

We have reviewed the literature (for example, W. A. Little, Phys. Rev. **134**, A1416 (1964)) and revised our discussions accordingly. We propose the following scenario: the electron phonon coupling is weakened in the beginning due to the abundant Se adatoms; With annealing, the strong e-ph coupling emerges, leading to the enhancement of superconductivity. The manuscript is revised as follows:

“Especially, phonons supported by these bonds in STO may interact with electrons in FeSe and raise the transition temperature. For example, the optical phonon mode responsible for the replica bands seen in ARPES¹² propagates along the Ti-O directions³⁴. At the as-grown state, the abundant Se adatoms may largely screen the dipole field generated by the F-K phonon modes in STO¹⁴ and the substrate mediated electron-phonon coupling poorly manifests itself. With annealing, Se adatoms desorb and the electric field can penetrate to the FeSe such that the substrate mediated electron-phonon coupling becomes stronger.”

5. Page 5 last paragraph This paragraph is unclear and should be made more precise.

a) The notation $p_{1/2}$ and $p_{3/2}$ and $5/2$ is not introduced

b) I failed to understand the logic of the two sentences “The Fe 2p spectrum ... ” until “ as has been observed previously in bulk FeSe and CeFeAsOF”

c) “... those of Fe in the zero valence state ...” The authors should explain what is meant by “the zero” in this case.

We have rewritten this part accordingly: “The Fe 2p spectra consist of $2p_{3/2}$ and $2p_{1/2}$ peaks due to spin-orbit interactions. As a reference, Fe foil corresponds to zero valence state: Fe⁰ (dotted black curve in Fig. 3a). The resemblance of the spectrum from the 20uc-FeSe/STO (solid black curve in Fig. 3a) to that of Fe⁰ reflects the delocalized nature of electrons in the Fe 3d band, as has been observed previously in bulk FeSe²⁹ and CeFeAsO_{0.89}F_{0.11}³⁹.”

6. Page 6 first paragraph: “Screening around Fe ... enhances the correlation effects of Fe bands only. A recent calculation ... of the balance states” Before this sentence the authors are

referring to the Fe p state and I believe they want to make the claim that the 2p state moves up in energy because of correlations that increase the effective mass and not because the Fermi level is reduced. However, they also say that from this peak the doped electrons concentrate on the Fe layer and I don't see who these conclusions are arrived at. I think the entire paragraph should be reformulated to state clearly 1) what is measured 2) the hypothesis and 3) why the data allows for this interpretation.

In the previous manuscript, we tried to provide two possible mechanisms for the contrasting behaviors between the Fe 2p peak and the Se 3d peak. The first one is from the correlation effect which renormalizes the Fe 2p orbitals. The second one is more simple-minded. From an atomic point of view, the binding energy of the core level states decreases if more electrons surround this atom, i.e. the atom tends to lose electrons. We therefore tried to interpret the lowering of binding energy in Fe only as an indication that electrons tend to concentrate on Fe.

As the referee pointed out, this section might cause some confusions. We have therefore removed the second explanation and rewritten it as follows: "Similar to the evolution of Fe 3d peak revealed by UPS (Fig. 1c), the change in Fe 2p peak also saturates once entering the superconducting regime, providing evidence for the correlation effects on the core level states. This finding expands the previous ARPES studies focusing on the energy range close to the Fermi level^{30,31}."

7. Page 6 second paragraph: "... the barrier across the FeSe/STO junction becomes smaller and the wavefunction of FeSe electrons spread deeper into STO ..."

While I can understand this argument from a particle-in-a box like argument, the Fermi level electrons are mostly Fe which is relatively far away from STO. Most importantly, for electron phonon coupling (and this is where the authors are going for) the vibration of an atom has to induce a potential variation for the particular electronic state in question. Even if Fe electrons tunnel into the STO more with annealing, it is unclear why this would increase the electron phonon coupling because the Fe and the Ti-O bond do not hybridize. The model put forward by the original forward-scattering nature paper is in disagreement with this picture, because they have an effective distance d of the polar mode in STO and the Fe layer and a large d ensures the sharpness of the coupling parameter $g(q) \sim \delta(q)$ required to explain the replica bands. Thus, the logic the authors present in this paragraph is not sound. If they want to make a

statement regarding electron-phonon coupling they should embed their picture and data into the relevant literature. For example, following their picture of increased tunneling of Fe into STO, I would have to expect that the electron-phonon coupling becomes broader in q which is not in line with the observed replica bands and in contrast to the claim of the paper.

We thank the referee for providing detailed explanation on the substrate mediated electron-phonon coupling. The permeation of electrons is, however, not essential in our interpretation of the mechanism for the enhanced electron-phonon coupling. We have removed the discussion about the permeation of electrons as has been stated in the answer to point 4.

8. Page 6 and 7, last paragraph: “Our findings here echo ...” The authors want to make the reader aware of their picture of the cuprates, but I feel the sentence is out of place, because it is unclear why a similar model in an unrelated material should help to understand the present case. If they want to keep a similar sentence, the authors should explain clearly in what sense the cuprates are related to FeSe on STO and thus, why the analogy helps to understand the present material.

9. Page 7: “Novel superconductors may be ...” The authors conclude with two sentences how novel superconductors could be constructed. Their suggestions are vague and largely based on the picture presented with the enhanced penetration of Fe electrons into the STO. As mentioned before, this picture is not in line with the literature and also not argued carefully.

We combine the answers to point 8 and 9 together since they are related.

We agree that the discussion in this paragraph is vague. We have decided to remove direct comparison with cuprates. We have used the following text as a replacement: “This work sheds light on choosing the appropriate material for realizing interface enhanced superconductivity. Apart from the demanding growth conditions such as matching the lattices, the substrate should have lower (higher) work function to allow for charge transfer that promotes the charge carrier density in the n-type (p-type) charge-transporting layer and have high energy phonon modes to further enhance the pairing strength.” .

Reviewer #2 (Remarks to the Author)

In the present work, the authors report STM, UPS, and XPS results on monolayer FeSe/STO. The authors performed very careful UPS and XPS measurements on the well characterized FeSe/STO with superconducting gap confirmed by STS. Such conventional UPS and XPS on the well-defined samples can be more useful than high-resolution ARPES for understanding of interface electronic structure. Indeed, this is the case in the present study.

The evolution of Fe 3d band with annealing is very interesting and consistent with the emergence of superconductivity. Also the decrease of band bending and work function with annealing is clearly shown by the systematic UPS and XPS data and supporting the band profile model.

1. I suspect that the popup in the O 2p band would be related to a kind of disorder in the topmost FeSe layer since the intensity of Fe 3d band is reduced in the 500 °C annealed sample (Fig. 1b) and would be consistent with the increase of exposed STO area in Fig. 1a.

The referee's suspicion is understandable, giving the strong Ti-O bonding peak observed in pristine STO. In the modified manuscript, we have spent more words on discussing this particular temperature point at 500 °C. In order to highlight the possible contribution from the exposed STO, we now use a lighter color for the data points at 500 °C. However, we find that such a popup still exists even after subtracting the contribution from the exposed STO. Detailed calculations have been added in the Supplementary Fig. 3. We reason that at the exposure percentage of 14% (500 °C) the contribution from the exposed STO is still minor: The covered STO contribute to the measured intensity of spectrum in the amount of 86% (coverage ratio) times the damping factor due to the presence of 1uc-FeSe. We estimate this damping factor to be 54% based on the realistic distance from FeSe to STO (0.61 nm from F. Li, et al. 2D Mater. **3**, 024002 (2016)) and the electron mean free path of 1 nm at this ionization energy. The covered STO therefore still dominates the measured spectrum in the energy range of O 2p derived features. The seemingly reduced intensity of the Fe 3d peak at 500 °C in Fig. 1b may come from both the reduced coverage and the relatively enhanced O 2p derived features, as we normalize our spectrum by the integrated intensity in the energy range of the O 2p derived features.

2. I recommend the authors to show Fe 2p XPS spectra (as well as Se 3d and Sr 3d) for the 500 °C annealed sample in Figs. 2 and 3 and to reexamine the electronic states of the 500 °C annealed sample.

We have included the 500 °C data of XPS for Fe, Se in Fig. 3 and Sr in Fig. 2. In the supplementary information, we have added a figure about the Fe spectrum and discussed the full width at half maximum (FWHM) of the Fe 2p_{3/2} peak (Supplementary Fig. 8). The peak width (FWHM) even shrinks a bit, again reflecting the renormalization effect. We do not see any dramatic broadening of the peak. It seems to suggest good crystalline quality of FeSe at 500 °C. In addition, we have reproduced the same increasing I_b/I_{nb} ratio on a second sample.

Reviewer #3 (Remarks to the Author):

The manuscript “Origin of charge transfer and enhanced electron-phonon coupling in single unit-cell FeSe films on SrTiO₃” by Zhang et al. presents UPS/XPS studies on the high T_c superconductor 1uc FeSe/STO. The origin of the enhanced superconductivity in this system has been a popular research topic, because the answer might help to guide the design and research of other interfacial superconductors. This manuscript measured the enhanced Ti-O bonding and the evolution of band bending during the annealing process from NS to S states. It also discussed the effect on charge transfer and e-p coupling. The paper is well written and seems scientifically sound. I want authors to clarify the following questions and provide some discussions in the revised version before the manuscript can be published in Nature Communications.

1. Please provide error bars for Figures 1 and 2.

We have added the error bars accordingly in Fig. 1c, 2b and 2d. “Error bars of the peak positions were estimated by considering the energy resolution of the measurement (0.05 eV).”

2. The authors show that the Ti-O bonding becomes weak once the FeSe deposited and stronger during the annealing process. What’s the I_b / I_{nb} ratio for annealed STO substrates? If the bonding in FeSe/STO after annealing is similar to STO, the authors should be careful about claiming the bonding is “strengthened” (compared to STO). The deposition of FeSe may weaken the Ti-O bond on the surface at the beginning, and later it becomes bulk-like during the annealing.

We have added the I_b/I_{nb} ratio for pristine STO in Fig. 1d. In fact, from the data taken with He-I light, I_b/I_{nb} seems to recover to its original value in STO. Interestingly, the data points taken with He-II show otherwise: I_b/I_{nb} can exceed the initial ratio of STO, suggesting that with annealing the Ti-O bond becomes even stronger than that of pristine STO. However, we remain cautious on drawing conclusion from the data from He-II alone. Understanding the discrepancy mentioned above may require measurements at different emission angles.

Taken the referee’s advice, we focus instead more on the evolution of I_b/I_{nb} with FeSe/STO changing from non-superconducting to superconducting and try to avoid the direct comparison

to pristine STO. We have added explicitly: “in comparison to the as-grown state” to the necessary places. In addition, we wish to note that we have reproduced the same increasing trend on a second sample.

3. Several papers have shown the double TiO₂ layer on STO surface (2D Mater., 024002, 2016 (Ref. 34 in the text) and PRB, 195303, 2016 (not in Ref.), and so on). The measured Ti-O bond length is not obviously smaller in the 1st TiO₂ layer. Also the bottom Se-Ti distance is bigger than expected, which is shown to be the opposite in calculations PRB, 220503, 2013 (Ref. 21 in the text) as the result of enhanced coupling of FeSe and STO. Can the authors provide some discussions?

We thank the referee for raising such an interesting point. We have added reference PRB **93**, 180506 (2016) from the same group that published PRB **93**, 195303 (2016) as Ref. 37 in the revised version. This reference is more related to our discussions.

In our previous simulation, we used a standard structure of STO with single TiO₂ on the surface. We also assumed the same bending values for TiO₂ and SrO if they are in the same unit cell. To check the possible influence of the double TiO₂ structure, we have improved our simulation code over the two above mentioned points. Details are described in the supplementary information.

We have updated Fig. 2 **c d** according to our improved simulation. In the main text, we consider the realistic situation of a double TiO₂ structure. We also note that the difference between the fitted results from this realistic structure and a standard single TiO₂ one is smaller than the energy resolution of our instrument. This discussion is added in the supplementary information: Supplementary Fig. 5.

Reviewers' comments:

Reviewer #1 (Remarks to the Author):

I think that the authors have met the points that I raised in my previous report.

The paper is now clearly written and I recommend publication in the current form.

Reviewer #2 (Remarks to the Author):

I found that most of the revisions and the responses are satisfactory and that the revised manuscript is probably publishable. However, I still have some concern on the interpretation of the popup in the O 2p band. Here, let me provide an optional suggestion.

The authors estimate the damping factor of photoelectrons from covered STO based on the distance from FeSe to STO (0.61 nm) and the electron mean free path of 1 nm at this ionization energy. Based on the damping factor, the authors conclude that the O 2p signal is dominated by the covered STO. On the other hand, in the revised version, the authors claim that "At the as-grown state, the abundant Se adatoms may largely screen the dipole field generated by the F-K phonon modes in STO14 and the substrate mediated electron-phonon coupling poorly manifests itself." If the abundant Se adatoms exist at the surface, photoelectrons from the STO substrate would be reduced further. In that case, the estimation of the damping factor should be reconsidered at least for the low temperature annealed samples. If possible, the authors should provide more quantitative estimation of the amount of Se adatoms and their effects on screening of the electric field as well as on damping of the photoelectrons. I suspect that the contribution from the covered STO is much smaller than the authors' estimation for the low temperature annealed samples. Also in Fig. 3, the Se 3d signal is reduced in the 500 oC annealed sample suggesting a drastic change between 480 oC and 500 oC annealed samples. Without careful and quantitative analyses, it is dangerous to assign the popup to the strengthened Ti-O bond.

Reply to the reviewer:

Reviewer #2 (Remarks to the Author):

I found that most of the revisions and the responses are satisfactory and that the revised manuscript is probably publishable. However, I still have some concern on the interpretation of the popup in the O 2p band. Here, let me provide an optional suggestion.

The authors estimate the damping factor of photoelectrons from covered STO based on the distance from FeSe to STO (0.61 nm) and the electron mean free path of 1 nm at this ionization energy. Based on the damping factor, the authors conclude that the O 2p signal is dominated by the covered STO. On the other hand, in the revised version, the authors claim that "At the as-grown state, the abundant Se adatoms may largely screen the dipole field generated by the F-K phonon modes in STO14 and the substrate mediated electron-phonon coupling poorly manifests itself." If the abundant Se adatoms exist at the surface, photoelectrons from the STO substrate would be reduced further. In that case, the estimation of the damping factor should be reconsidered at least for the low temperature annealed samples. If possible, the authors should provide more quantitative estimation of the amount of Se adatoms and their effects on screening of the electric field as well as on damping of the photoelectrons. I suspect that the contribution from the covered STO is much smaller than the authors' estimation for the low temperature annealed samples. Also in Fig. 3, the Se 3d signal is reduced in the 500 oC annealed sample suggesting a drastic change between 480 oC and 500 oC annealed samples. Without careful and quantitative analyses, it is dangerous to assign the popup to the strengthened Ti-O bond.

We thank the referee for spending efforts on finding an alternative, which leads us to further refinement. Below we list our points, hoping that they answer the referee's concern:

(1) We estimate the attenuation of photoelectrons from the STO base on the formula:

$$\gamma = \exp\left(-\frac{d}{\lambda}\right),$$

where d is the distance from FeSe to the top surface of STO. λ is the mean free path of electrons. As the referee pointed out, for FeSe with abundant Se atoms, the mean free path may be shorter than the value in stoichiometric FeSe. However, the value of 1 nm is already an underestimation. On the "universal curve" (Surf. Interface Anal. 1, 2 (1979) and reference [3] in the supplementary information), there exists a certain scattering of the data points from real materials (we attached their figure for convenience). Our estimation, by using a kinetic energy of 15 eV (photon energy 21.22 eV- binding energy 6 eV), actually falls on the lower bound of that distribution. In fact, an optimistic value of λ would be 3 to 5 nm. In that case, γ exceeds 0.8, meaning that most of the photoelectrons can pass through.

(2) We observe that the Ti-O bonding peak becomes prominent "relative to" the Ti-O nonbonding peak. We avoid judging from the absolute values of peaks (in both regimes) as the normalization factor may vary. For the convenience of discussion, we wish to separate the annealing temperatures into two regimes:

regime-I: 300°C, 350°C, 400°C. The surface is 100% covered by 1uc FeSe, and the extra Se adatoms decrease in amount with successive annealing;

regime-II: 450°C, 480°C, 500°C. There are basically without Se adatoms, while the coverage of 1uc-FeSe decreases slightly with successive annealing.

In regime-I, The damping factor, as the referee pointed out, may be different from the value estimated from a stoichiometric 1uc-FeSe. Nevertheless, this damping factor should have a weak energy dependence. We believe that it is very unlikely that the Se adatoms damp particularly strongly at the energy of the Ti-O bonding peak, leaving the Ti-O nonbonding peak which is just 2 eV apart undamped. In other words, Se adatoms should provide a uniform damping across the energy range we studied. Dividing the Ti-O bonding peak by the nonbonding peak then cancel out the damping factors in both the numerator and the denominator.

In regime-II, however, the damping factors cannot be canceled out by dividing the peak intensities of the two because there exists some exposed STO where the photoelectrons are not damped. It is in this regime that we argue the signal from the covered STO still dominates. We wish to point out that, in this regime the damping factor resumes to be 0.54 since there is no more Se adatoms. The apparent drop in intensity of the Se 3d peak reflects both this reduction and the incomplete coverage. The pop-up in this regime therefore cannot be explained by the suggested scenario of extra Se adatoms.

Our intension of displaying Fig. 3 is to demonstrate the shift of the peak position rather than the absolute intensities of the peaks. In order to avoid confusing the readers, we have refined our normalization method by first subtracting the background (at 50 eV) and then normalize the spectra by the integrated intensity of the subtracted spectra in the energy range from 50 to 60 eV. By doing so, we removed any obvious change in the peak intensities. The revised Figure 3 has been attached.

Figure 3 Fe 2p and Se 3d core-level spectra of 1uc-FeSe/STO. Bottom curves in the left panel of **a** (the main panel of **b**) are reference spectra of 20uc-FeSe/STO and a Fe foil (a 20 nm thick Se film). The Fe foil has been repeatedly sputtered and annealed to ensure the removal of oxidized layers. **c**, Summary of the peak positions of Fe $2p_{3/2}$ and Se 3d at different annealing stages. The peak positions of Fe $2p_{3/2}$ are extracted from the Gaussian-Lorentzian fitting after subtracting the Shirley background (see Supplementary Fig. 8). The peak positions of Se 3d are obtained from the corresponding energy values of the intensity maxima.